# Early Crop Classification via Multi-Modal Satellite Data Fusion and Temporal Attention

**Frank Weilandt [1], Robert Behling [2], Romulo Goncalves [2], Arash Madadi [2], Lorenz Richter [1,3], Tiago Sanona [1], Daniel Spengler [2]** and **Jona Welsch [1,*]**

1   dida Datenschmiede GmbH, Hauptstr. 8, Meisenbach Höfe, 10827 Berlin, Germany
2   Helmholtz-Zentrum Potsdam, Deutsches GeoForschungsZentrum GFZ Telegrafenberg, 14473 Potsdam, Germany
3   Zuse Institute Berlin, Takustraße 7, 14195 Berlin, Germany
\*   Correspondence: jona.welsch@dida.do

**Abstract:** In this article, we propose a deep learning-based algorithm for the classification of crop types from Sentinel-1 and Sentinel-2 time series data which is based on the celebrated transformer architecture. Crucially, we enable our algorithm to do early classification, i.e., predict crop types at arbitrary time points early in the year with a single trained model (progressive intra-season classification). Such early season predictions are of practical relevance for instance for yield forecasts or the modeling of agricultural water balances, therefore being important for the public as well as the private sector. Furthermore, we improve the mechanism of combining different data sources for the prediction task, allowing for both optical and radar data as inputs (multi-modal data fusion) without the need for temporal interpolation. We can demonstrate the effectiveness of our approach on an extensive data set from three federal states of Germany reaching an average F1 score of 0.92 using data of a complete growing season to predict the eight most important crop types and an F1 score above 0.8 when doing early classification at least one month before harvest time. In carefully chosen experiments, we can show that our model generalizes well in time and space.

**Keywords:** early crop-type classification; satellite image time series; Sentinel-1; Sentinel-2; multi-modal remote sensing data; deep neural networks; temporal attention encoder; data fusion

## 1. Introduction

The ongoing climate change, a rising world population, the increasing use of agricultural products as energy sources, as well as a growing demand for the integration of sustainable cultivation forms are evident major challenges for the agriculture of the future. For all those aspects, access to detailed area-based information is crucial, e.g., for the creation of yield forecasts, agricultural water balance models, remote sensing-based derivation of biophysical parameters or precision farming. Of particular relevance is the knowledge and monitoring of crop types. It is highly relevant for a number of user groups, such as public institutions (subsidy control, statistics) or private actors (farmers, agro-pharmaceutical companies, dependent industries). For most use cases, in fact, the respective crop type information is usually needed as early as possible, ideally several months before harvest.

The identification of crop types has a long research history in remote sensing [1–5]. In the 1980s it started by using (un-)supervised classification approaches using single optical images [3,4,6]. Meanwhile, studies have shown that crop-type classification benefits from integrating remote sensing time series data instead of mono-temporal information [3,4], combining complementary information of optical and radar data instead of single sensor approaches [2,7–11], as well as from utilizing deep learning models in favor of classical supervised approaches [4,12]. While during the last decade decision-based models, such as random forests, were commonly used for crop type classification [13–17], subsequently deep learning models, such as convolutional, as well as recurrent neural networks, proved

to be very successful [18–20]. More recently, experimental comparisons showed that the self-attention layer as used in Transformer models [21] yields even more precise results for crop type prediction on remote sensing data [22,23].

However, besides the recent methodological advancements, crop types are still often identified only retrospectively after harvest or in rather late cultivation phases. Additionally, most studies have been developed on relatively small reference datasets [2], which is most likely also one of the main reasons why the transferability of the approaches in time and space has been hardly addressed so far [2]. Larger reference data sets are required to exploit the full potential of deep learning approaches and to develop models that generalize well.

Another aspect concerning input data is the fusion of the multi-modal information of optical and radar remote sensing data. So far, this has been realized at different stages of the algorithm [2,24,25] and a straightforward method for data integration of spatiotemporal heterogeneous data coverage of different modality keeps being a major challenge for crop-type classification approaches.

In this article, we aim to address those issues by proposing an approach for progressive intra-season crop-type classification with a seamless integration of spatiotemporal data sources. Our developments are mainly based on the self-attention-based architecture developed by [22], which has proven to work well in the classification of crop types in satellite time series data. However, certain methodological developments are necessary in order to allow for the real world scenario of an ongoing near real-time crop classification as the growing season progresses. In particular, the model needs to be able to reliably predict crop types in time series data of arbitrary length in a year the model was not trained on. Thus, our study evaluates different strategies for early crop type classification and focuses on the ability to be transferred to unseen data in time (different years) and space (different regions). Moreover, the goal of large area crop classification requires the model to be flexible in terms of the input data type in order to ensure reliable data availability for each region and time the model is applied to. Hence, the second large focus of our work is on developing seamless data fusion techniques which shall especially be able to handle the freely available dense time series of Sentinel-1 (radar) and Sentinel-2 (optical) data. It has already been shown that using both sensors simultaneously improves prediction performance [24,25], however, the progressive early crop classification brings some special requirements also for the data-fusion techniques. Therefore, the data fusions need to be flexible and, in particular, be applicable to spatiotemporally inconsistent time series data of arbitrary length.

The development of our approach is based on crop data from three German federal states (Brandenburg, Mecklenburg-Vorpommern, and Thuringia) from 2018 until 2020 comprising more than 1.3 million reference parcels (i.e., agricultural fields). This large and heterogeneous reference dataset allows for a well-balanced training and testing of the approach in different scenarios.

The main contributions of our work can be summarized as follows:

- We propose and implement a deep learning-based method for predicting crop types during the entire year, in particular allowing for early classification.
- We systematically compare model performances in different cultivation stages (from early to late season).
- We present a generic way of combining input data from several different sources, in particular focusing on the fusion of data from Sentinel-1 and Sentinel-2 satellites.
- We predict crop types in a year that is not available during training (temporal transfer).
- We apply our method for the prediction in another region (spatial transfer).

The article is structured as follows. Section 2 gives an overview of the used datasets and lists the most relevant crop types. Section 3 summarizes the machine learning model and highlights our methodological improvements. Section 4 shows the results of multiple experiments that evaluate different aspects and scenarios of the approach. It follows a step-wise procedure to support methodological decisions made towards the real world scenario of progressive early crop-type classification. The performance of the approach and

its methodological aspects are discussed in the context of the relevant literature in Section 5, followed by a final conclusion of the findings in Section 6.

## 2. Data

For this study, parcel boundary information is combined with Sentinel-1/-2 data for three German federal states (Brandenburg, Mecklenburg-Vorpommern, and Thuringia), shown in Figure 1, and three years (2018, 2019, and 2020). The regions have been chosen because of the large field sizes, therefore having high potential and relevance for large scale remote sensing based analysis. Furthermore the crop data for training and validation were openly accessible or have been provided by the federal states. The diversity of data is crucial for showing the temporal and spatial generalization of our models and especially for showing the feasibility of our methods for early classification of crops in a year not appearing in the training data.

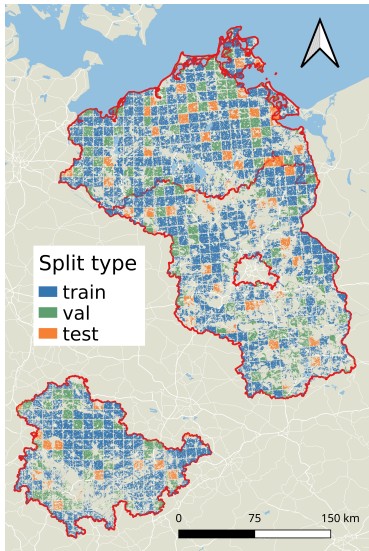

**Figure 1.** Areas which are used for training, validation, and test. Colors correspond to the split to which parcels were assigned. The assignment is made based on 10 km MGRS tiles, as described in Section 3.5. Each tile contains multiple parcels. The red lines delineate the state boundaries. Mecklenburg-Vorpommern is the topmost highlighted state, Brandenburg is below and Thuringia is the isolated shape on the lower left corner. Map data from OpenStreetMap [26] and state boundary polygons from German Federal Agency for Cartography and Geodesy [27].

### 2.1. Satellite Data

Observations from the Sentinel-1 (S1) and Sentinel-2 (S2) satellites from the Copernicus program are used.

**Sentinel-1:** In case of S1 backscatter coefficients (sigma nought) in decibel (dB) of two polarizations (VH, VV) from the S1 SLC product are used. The data are processed by using pyroSAR [28,29] to apply SNAP [30] functionalities in a pipeline (i.e., "Apply-Orbit-File", "ThermalNoiseRemoval", "Calibration", "TOPSAR-Deburst", "Multilook", "Speckle-Filter", "Terrain-Correction"). During terrain correction the S1 data are aligned to the S2 data in terms of the same spatial resolution (10 m), the same map projection, and the same pixel grid. Data of the ascending orbit are used (approx. sensing time of 5 pm) to have a consistent data base and to avoid more frequent dew and rime in earlier sensing times of the descending orbit (approx. sensing time of 5 a.m.) [31].

**Sentinel-2:** S2 data are provided as bottom of atmosphere Level2A data products. The atmospheric correction is performed using Scicor [32]. Scicor enables the classification of cloud conditions and a classification into broad surface categories [33]. Pixels representing reflectances of Earth's surface under clear sky conditions are used for further analysis.

Pixels classified by Scicor as cloud, cirrus, shadow, snow, or water are excluded and masked as no-data pixels. We use all the 10 m and 20 m bands of S2 (B2, B3, B4, B5, B6, B7, B8, B8A, B11, and B12) in their original spatial resolution.

We use data from within a hydrological year to reflect the possible phenological stages of agricultural crops. For example, when referring to the 2019 data, we use the observations acquired on days from 1 October 2018, to 30 September 2019. The acquisition dates differ between parcels and sources (for example due to cloud obstruction). This is shown for two exemplary parcels in Figure 2.

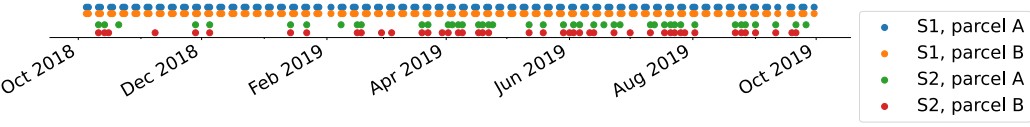

**Figure 2.** Temporal availability of S1 and S2 data for two example parcels from Brandenburg for the 2018–2019 growing season. The S1 and S2 satellites usually do not acquire observations for a parcel on the same days. The temporal pattern of S2 observations is irregular, because we drop the acquisitions when the parcel is occluded by clouds. Therefore, parcel B has for example more S2 acquisitions in October 2018 and March 2019 than parcel A.

### 2.2. Reference Data for Crop Types and Regions

Reference data for training and validation are based on the Integrated Administration and Control System (IACS), providing the crop type information. Data sets from three federal states of Germany, Brandenburg (BB), Mecklenburg-Vorpommern (MV), and Thuringia (TH) have been used for this study. Together, they contain information for over 1.3 million reference parcels. The data for Brandenburg are freely available [34], the data for the other two states were provided to the authors by the corresponding authorities upon request. They are based on Shapefiles which contain polygons outlining parcels and the respective crops that farmers have declared to their agricultural administrations to be growing there for each year and state. Since each federal state has its own scheme for distinguishing crop types, we harmonized the labels using the definitions from the Joint Experiment for Crop Assessment and Monitoring (JECAM), as described in [35] (for details about this harmonization see Appendix F). This way, we use clear crop type definitions, allowing for the extension of the labeling scheme to other German federal states or regions in other countries.

All parcels appearing in these declarations are used, except very small parcels that do not cover at least one Sentinel-1/-2 pixel. We describe crop classes with the following four kinds of crop types:

- *Main crop types*: maize, potatoes, rapeseed, sugar beet, winter barley, winter rye, winter wheat;
- *Additional crop types*: asparagus, lupins, meadows, peas, spring barley, spring oats, spring rye, spring wheat, sunflower;
- *Not considered*: any class not listed above.
- *Other*: *additional crop types* and *not considered*.

The main crop types are the most relevant crop types in our study area. The selection has been based on statistical appearance, economic impact, and also the potential of classification for class separation, based on remote sensing expertise. We train our models on all crop types, but usually report only the metrics on the main crop types to keep the presentation concise. More precisely, we will distinguish the seven main crop types and the additional class *other*, comprising all the crop types which are not among the main crop types (including the crop types which are not considered).

We show the amount of parcels per crop type in our dataset in Figure 3 for all three years 2018, 2019, and 2020. These numbers do not vary much between the years for a given state, but they can vary between states. For example, asparagus is more common in

Brandenburg than in the other states, and winter rye and lupins are comparatively rare in TH.

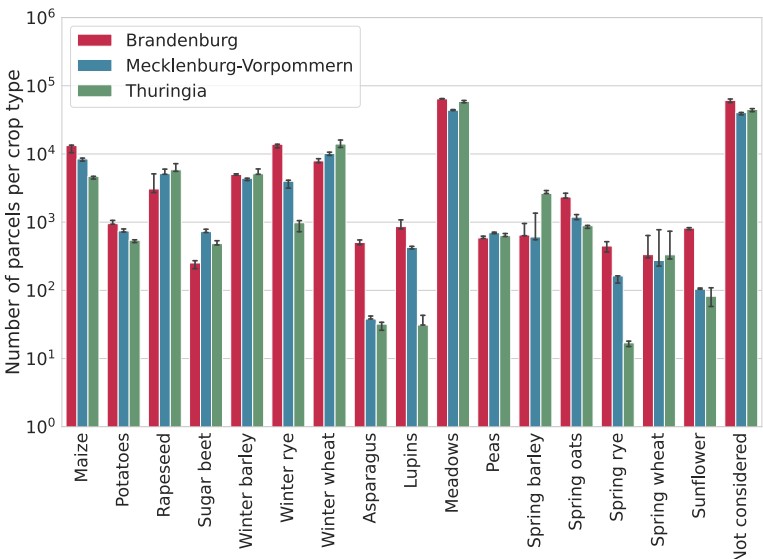

**Figure 3.** Amount of parcels per crop type in different years and different states in absolute numbers. For each state and crop type, the height of the bar represents the number of parcels with this crop type of the median year, i.e., the upper edge of the error bar shows the maximum number during the three years, the lower edge the minimum number and the height of the bar the intermediate (median) number within the years.

## 3. Methods

In this section, we detail the developed methods and in particular elaborate on the used machine learning model and its early classification, as well as data fusion capabilities. The model builds on previous work [22] and leverages the *attention mechanism* originally introduced in the famous Transformer architecture [21], aiming to better exploit relevant information about crop types included in the change of satellite images over time. At the same time, certain aspects need to be modified, such as, e.g., the positional encoding, in order to be able to train and predict on regions and data sources with differing acquisition time points.

### 3.1. Deep Learning Model

Our work mainly builds on the Pixel-Set Encoder–Temporal-Attention Encoder (PSE-TAE) model introduced in [22]. This neural network consists of three steps which are executed for each parcel. These steps and the most important modifications to them in this study are briefly outlined in the following (see also Figure 4):

1.  Pixel-set encoder (PSE): fully connected neural networks are applied to a random sample of 10 pixel positions from a parcel. This yields a vector $e^{(t)}$ for each index $t \in \{1, \ldots, T\}$ (all input time series are padded to a common length of $T = 140$). The pixel sampling is random during training, i.e., usually a different set of pixels is chosen every time a parcel is loaded. However, the random number generator is seeded during evaluation to make the predictions for a given model reproducible. We also experiment with a different encoding technique: we simply pass the averages of the selected pixels to the TAE (see Appendix D).

2.  Positional encoding: time information $p^{(t)}$ is added to each vector $e^{(t)}$. The positional encoding mechanism thereby takes care of different temporal acquisition patterns, as it adds the information about the acquisition date directly to the parcel observation serving as an input for the next step. The specific implementation of the temporal encoding is the most important architectural change in this study compared to [22]:

In contrast to the original model, the actual acquisition day relative to the start of the growing season (e.g., "day 6") instead of the acquisition count (e.g., "acquisition number 2") is used to calculate $p^{(t)}$. This provides greater flexibility with regard to the temporal patterns in the input data (see also Section 3.3). More details about the positional encoding mechanism can be found in Appendix A.2.

3.  Temporal-attention encoder (TAE): the main component of this step is the attention mechanism originally introduced in the famous Transformer architecture [21] for natural language processing (NLP). There it is used to encode text as a sequence of vectors indexed by the position of each word in a sentence and specifically encourages interactions between different words, i.e., it can learn context. This allows to compute semantically meaningful representations of sentences. Based on the observation that a parcel's development can be represented as a sequence of vectors indexed by time and that the change of a parcels appearance over time is an important factor when classifying crop types, attention is used in an analogous setting here: instead of a sequence of words, the corresponding neural network layer encodes a time series of parcel observations and their interactions, which is then used to compute the final classification.

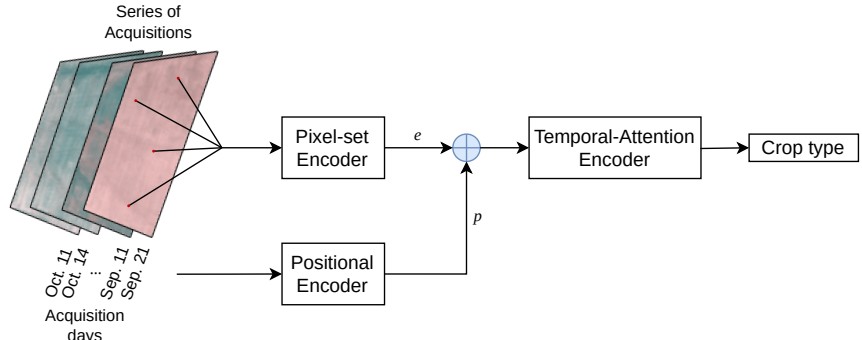

**Figure 4.** Sketch of PSE-TAE architecture. In this example, $\text{day}(1) = 10$ because the first observation was acquired 10 days after October 1st.

Further details on our model and modifications with respect to [22] can be found in Appendix A, a description of our training parameters can be found in Appendix C.

### 3.2. Normalization

Since we want to concatenate data from Sentinel-1 (S1) and Sentinel-2 (S2), first a normalization of the input data for each source needs to be completed. This usually helps a deep learning model to converge better. For each channel $c$, we collected the values observed in the train set for Brandenburg and Mecklenburg-Vorpommern (BB + MV) for 2019 and computed their mean $\mu_c$ and standard deviation $\sigma_c$. During data loading, each input channel $x_c$ is then normalized via $(x_c - \mu_c)/\sigma_c$.

### 3.3. Data Fusion

We deal with the varying resolution of satellite bands by converting everything to 20 m resolution. For data in 10 m resolution, we only use every other pixel along the horizontal and the vertical axis. Since we do not use texture but a set of sampled pixels, we do not expect to lose relevant information by this simple and efficient strategy. We did not try other strategies, but refer to studies such as [36,37] for alternatives. Note that the model itself is resolution agnostic: as we sample pixels for each parcel to be classified, it could be used with data in 10 m resolution without any changes.

We fuse the data early at the input-level before sending it to the model: this means that we concatenate the observations from S1 and S2 for each day on which there was an image acquisition from at least one of the two (concatenation along channels). For days on which there was only an acquisition from one of the data sources, the special

no data value NA $= -1$ is used to fill in the missing data from the other source. Since the positional encoding includes information about the exact acquisition day for each observation, no temporal interpolation is needed to align the time series from different modalities, regardless of vastly different acquisition patterns.

### 3.4. Early Classification

The main objective and use case is to predict crop types early during the year. Additionally, the application of a model trained on data from earlier years on a later year, from which no data have been seen during training will be performed. Crucially, only the input data are modified in order to learn early classification, the architecture described in Section 3.1 remains unchanged.

**Temporal masking:** the operation used for input data modification is the following: for a given satellite image time series $x$ and a date index $D \in \{1, \dots, 366\}$, we define

$$x[: D] = (x^{(1)}, \dots, x^{(t)}),$$

where $x^{(t)}$ is the last observation in $x$ acquired at most $D$ days after October 1st of the preceding year, i.e., $\text{day}(t) <= D$ and $\text{day}(t+1) > D$. We say the *time series $x$ bounded by the D-th day*. This means that we mask out some of the acquisition time points in order to simulate the situation in early crop classification when there is only data available up until a certain point in time. Specifically, this also means that the data fusion mechanism described in Section 3.3 can be applied in the same way as when classifying time series from a complete growing cycle.

We try two approaches for training models:

(S) Train *separate models* for separate periods, e.g., the model to be used at the end of May is trained on the time series $x[: 240]$ (end of May is approximately $8 \cdot 30$ days after 1 October). When predicting the crop type for a parcel with data available until a certain month, we use the corresponding model.

(U) Train a *unified model*: the training data spans full hydrological years. However, during training, we apply random temporal masking, i.e., we randomly select a day $D$ for each parcel and epoch and use only $x[: D]$ as input data. This way, we simulate having observations only until a certain date. Then we use this single trained model for all time series lengths at prediction time—no matter whether they contain only data from a few months or from the full year.

In the evaluation, we always use observations until a given date, e.g., $x[: 150]$ contains all observations until approximately the end of February (five months counting from October). Each model from approach (S) is evaluated with data restricted to the same period it was trained for. When evaluating approach (U) for each bound $D$ on the days, we simply evaluate the unified model on the data bounded by $D$. In practice, we run our evaluations for $D \in \{60, 90, \dots, 360\}$, see Section 4.3.

### 3.5. Train/Test Splits

The area of interest consisting of the three states is split into train, validation, and test regions. The area is covered by grid squares from the Military Grid Reference System (MGRS, Ref. [38]). We use the grid squares with precision 10 km as tiles of size 10 km $\times$ 10 km which cover the area, e.g., the MGRS coordinate 33UUV85 denotes such a tile lying inside Mecklenburg-Vorpommern. We created a random split of these tiles into train, validation, and test tiles—using a ratio of 7:2:1. Each parcel is assigned to train/validation/test depending on the MGRS tile covering it. We drop the parcels whose centers lay within a margin of 250 m to the tile border, in order to avoid data leakage between splits. For example, we assign the tile 33UUV85 to the train split, hence all parcels whose centers are located at least 250 m from the border of the tile belong to the train set. We show the result of this split for Brandenburg, Mecklenburg-Vorpommern, and

Thuringia in Figure 1. This split is kept the same during all years. We took the idea for this tile-wise split from [23].

### 3.6. Metrics

The dataset shows a strong class imbalance (Figure 3). Hence, measuring the prediction quality using overall accuracy would mean that the most frequent crop types would dominate the computation, for example winter wheat and the irrelevant class "other". We therefore decided to mainly report the (class-wise) F1 metric. However, we sometimes mention the overall accuracy as well. Mainly to make our results more easily comparable to related work, which often reports overall accuracy.

For every crop type, we count the number of correctly predicted parcels and incorrectly predicted parcels. This yields the usual recall (sometimes called producer's accuracy) and precision (user's accuracy). Their harmonic mean is the *F1 score* for each crop type.

We consider the arithmetic mean (macro average) of the F1 scores of the main crop types our standard measure of the prediction quality and call it *main F1 score*. Using this metric, each crop type has the same influence on the overall score. We always report metrics on the test set, the validation set is used for tuning hyperparameters during training and model development.

## 4. Results

The proposed methods are evaluated in a step-wise procedure to demonstrate their usability for varying crop type classification tasks and to support the decisions made in terms of methodological developments with the goal to reach the real world use case of near real-time progressive early classification. Therefore, each section consists of a description of the conducted experiment, depicts the model performances, and outlines the methodological decisions. Section 4.1 does a baseline model comparison to judge the performance of the proposed deep learning model (Section 3.1) against a classical machine learning algorithm, namely a random forest. Section 4.2 focuses on crop classification using data from a complete growing season to evaluate different aspects of our model, such as input modalities and the model transfer in time and space. The configuration that performed best is used in Section 4.3 to evaluate different early crop classification approaches and their transferability to current years to allow for ongoing intra-season crop type classification.

### 4.1. Baseline Model Comparison

For the baseline model comparison, a common and basic crop-type classification task is chosen. The compared methods classify crops using single-modal Sentinel-2 (S2) satellite time series data, which cover a single region during the complete growing season. The model performances are evaluated on the same year and region the model was trained on. As a baseline, we implemented a random forest (RF) model trained in a similar fashion to [16] with a few alterations. The model was trained on 500 trees and operates on S2 features (bands and spectral indices) commonly used for this type of task [13,17,39]. We trained this baseline model and a Pixel-Set Encoder–Temporal-Attention Encoder (PSE-TAE) model on data from Brandenburg and Mecklenburg-Vorpommern (BB + MV) for the year of 2018. Table 1 shows the results of evaluating both models using the test set of the previously mentioned dataset. We observe a clear advantage on using the deep learning (DL) model in favor of the classical model.

**Table 1.** Main F1 score for full year inference on test data using a RF and the DL model. Both models were trained and tested using data from BB + MV in 2018 and only using S2 observations.

|  | Main F1 |
| --- | --- |
| Random Forest | 0.72 |
| PSE-TAE | 0.91 |

We present a more extensive description of the baseline model, as well as crop-wise metrics in Appendix B.

### 4.2. Full Year Classification

By using data of the full growing season (i.e., hydrological year from October to September) the approach is evaluated in the following aspects. Section 4.2.1 investigates the model performances using different input modalities and evaluates the added value of using multi-modal data. Moreover, we analyze the capability of this algorithm to be transferred in time, i.e., predicting crop types for a year which was unavailable during training (Section 4.2.2), and in space, i.e., predicting crop types for a region which was unavailable during training (Section 4.2.3).

#### 4.2.1. Using Different Modalities

For the evaluation of the different input modalities the model is trained and tested on Brandenburg (BB) data from 2018. For each setting—Sentinel-1 (S1), Sentinel-2 (S2), or both (S1 + S2)—the training is repeated 5 times to make sure that potential effects are not due to the randomness of the model training. We evaluated each model on the test set and computed the average F1 score over all crop types (Table 2).

**Table 2.** Main and Overall F1 scores for full year inference on test data using S1, S2, or both. Each experiment was executed 5 times, means and standard deviations of the corresponding scores are given. All models were trained and tested using data from BB in 2018.

| Source | Main F1 | Overall F1 |
|:---:|:---:|:---:|
| S1 | $0.84 \pm 0.01$ | $0.66 \pm 0.01$ |
| S2 | $0.87 \pm 0.01$ | $0.72 \pm 0.001$ |
| S1 + S2 | $0.89 \pm 0.01$ | $0.76 \pm 0.01$ |

Table 2 shows that the PSE-TAE model trained with both modalities S1 and S2 usually leads to the highest F1 scores. The detailed results for each crop type can be found in Table A3, showing that the gain from fusing both data sources mainly appears in the crop types which are under-represented in the data.

#### 4.2.2. Temporal Transfer: Evaluation on Future Year

The best configuration of the model (using both: S1 + S2) is used to investigate its performance when predicting crop types in an unseen year. We trained a model on BB + MV and the years 2018 & 2019. To have a baseline, we first evaluated on the same regions and years, cf. Table 3 (metrics for all crop types can be found in Table A4). The crop-wise F1 scores are consistently above 0.9, except for potatoes.

To evaluate the generalization capability of this model it is applied to an unseen year (2020). Again crop-wise F1 scores consistently above 0.9 are reached, except for potatoes and winter rye (Table 3). The main F1 score here is 0.89 with a main overall accuracy of 96%. Hence, the main F1 score is only 0.03 lower than the main F1 score of 0.92 when evaluating on the training years of 2018 & 2019.

#### Different Combinations of Train/Evaluation Years

Since the weather conditions differ between the three years considered, we want to explore whether training on both years 2018 & 2019 is more helpful for the prediction on 2020 than training on only one of these years. Table 4 shows the main F1 scores for several combinations of years for training and evaluation.

**Table 3.** Comparison between evaluation F1 scores on years which were available during training and future year: evaluation on test data for BB + MV from 2018 & 2019 and 2020 using PSE-TAE with S1 + S2 as input. The model was trained on the training set for BB + MV from 2018 & 2019. The detailed metrics for all crop types are given in Tables A4 and A5.

|  | F1<br>2018 & 2019 | F1<br>2020 |
|---|---|---|
| Maize | 0.93 | 0.95 |
| Potatoes | 0.74 | 0.62 |
| Rapeseed | 0.97 | 0.97 |
| Sugar beet | 0.94 | 0.94 |
| Winter barley | 0.96 | 0.91 |
| Winter rye | 0.92 | 0.82 |
| Winter wheat | 0.94 | 0.91 |
| Other | 0.98 | 0.98 |
| Average \| Sum | 0.92 | 0.89 |

**Table 4.** The main F1 scores for different combinations of train/evaluation years. The score of the best-performing model for each evaluation year (column) is marked in bold. All models were trained and evaluated on the corresponding train and test sets from BB + MV.

| Train Year(s) | Year(s) Used during Evaluation | | | |
|---|---|---|---|---|
|  | 2018 | 2019 | 2018 & 2019 | 2020 |
| 2018 | 0.91 | 0.85 | 0.88 | 0.81 |
| 2019 | 0.75 | 0.92 | 0.85 | 0.86 |
| 2018 & 2019 | **0.92** | **0.93** | **0.92** | 0.89 |
| 2020 | 0.76 | 0.86 | 0.82 | **0.92** |

The comparison shows:

- When evaluating on a single year (2018 or 2019), the model profits from adding training data from another year.
- When evaluating on 2020, the model trained on earlier years 2018 and 2019 has a main F1 score 0.03 smaller than the model trained on the specific year 2020. Training on both earlier years (2018 & 2019) leads also to a better score than only training on a single earlier year (2018 or 2019).

4.2.3. Spatial Transfer to Thuringia

To test for spatial transferability the model trained on BB and/or MV is used to predict crop types in Thuringia (TH). The states BB and MV are located in the northeastern part of Germany, whereas TH lies in the center of Germany. Additionally, the topography in TH is hillier. Table 5 compares models trained in different regions and depicts the main F1 scores to predict crop types in TH.

**Table 5.** Main F1 scores for evaluating in TH and different combinations of train states. All models were trained and tested on the full years 2018 & 2019 on the corresponding training and test sets. Best score is marked in bold.

| Train State(s) | Evaluation in TH |
|---|---|
| BB | 0.80 |
| MV | 0.82 |
| BB + MV | 0.84 |
| TH | 0.86 |
| BB + MV + TH | **0.88** |

One can see that using data from more states leads to better results:

- When evaluating in TH, the model trained on the combination BB + MV reaches a main F1 score of 0.84, which is better than the models only trained on either BB or MV.
- The model trained on BB + MV (spatial transfer) reaches a performance similar to a model trained directly in TH, with a main F1 score of 0.84 and 0.86, respectively.
- Training on all three states (BB + MV + TH) further improves the performance in TH, compared to training on data only from TH.

*4.3. Early Classification*

Section 4.3.1 compares the two early classification strategies described in Section 3.4, using the best model configuration determined in Section 4.2. The best of these strategies is used to compare results between evaluation on the training (2018 & 2019) and the unseen (2020) years (Section 4.3.2), to evaluate the real world example of ongoing near-real time intra-season crop-type classification.

4.3.1. Comparing Approaches for Early Classification

The early classification strategies are the separate model approach (S) and the unified model approach (U), as described in Section 3.4. Like in Section 4.2.1, we train and test each of the model configurations five times and present means and standard deviations of our metrics over the runs.

Using approach (S), we trained five models for each of the seven time periods $x[:180]$, ..., $x[:360]$, corresponding roughly to the end of the months March to September. Additionally, five unified models using approach (U) were trained. These model trainings used the combined data from BB and MV during the years 2018 & 2019. We evaluated the models using the test sets of the same regions for both periods of 2018 & 2019 and 2020. To obtain F1 score values for different points in time we masked the time series data the same way as the training data of approach (S).

The average F1 score over the main crop types, obtained from evaluating each approach on either train years or the unseen year is plotted in Figure 5. We observe that both approaches perform similarly albeit with a small difference between March and the beginning of July for the evaluation on the training years. When looking at the performance over the unseen year, the curves seem to match most of the time, except in May and July. In Figure 5b, a drop in performance during the month of July can be observed.

Approach (S) has a slight advantage when inferring in the training years. Yet it is expensive to train a model essentially for every time point for which one would like to have predictions. In addition, approach (S) seems to perform very similarly to Approach (U) when using the unseen year. The later (train on reference years and inference on unseen years) would be the configuration used in a real life application of this model. Given these facts, we proceed with testing and presenting results for early classification using only the unified model approach (U).

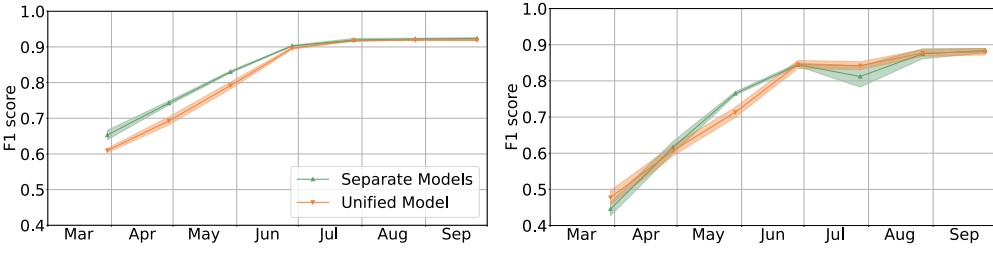

(**a**) Evaluation on training years 2018 & 2019      (**b**) Evaluation on unseen year 2020

**Figure 5.** Comparison of average F1 scores over the main crop types between approaches (S) (separate models) and (U) (unified model). Each vertical line represents the first day of a month, and each dot's horizontal position marks the last day whose acquisition was used during prediction. The colored bands represent the standard deviation of the scores over five training runs. The average F1 scores plotted here are given in Table A6.

### 4.3.2. Evaluation on Training Years vs. Unseen Year

The unified model (U) is used to compare early crop type classification metrics of models which are applied to the trained years (2018 & 2019) with models applied to an unseen year (2020). Figure 6 presents the results of a small selected area in northern Brandenburg for the classification year 2020. At selected fields the classification results are varying related to the selected end date of the classification. This development is aligned with the progressively increasing classification accuracy presented in Figure 7.

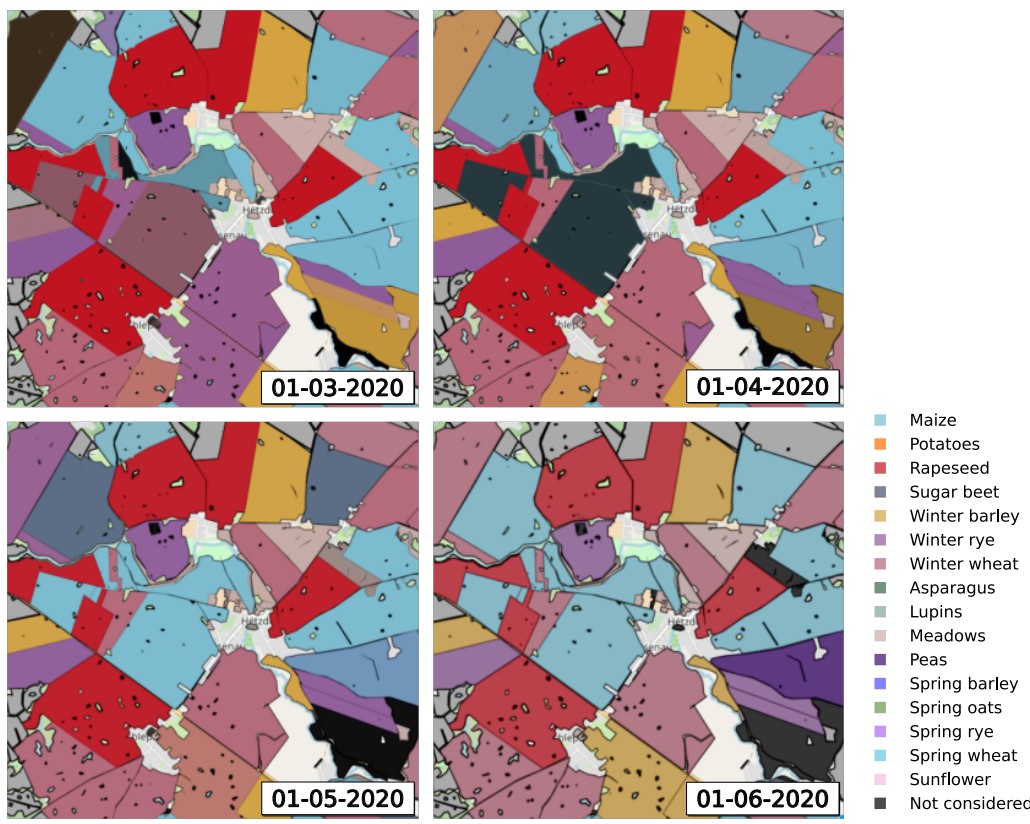

**Figure 6.** Classification results of a test site in northern Brandenburg for year 2020 at four different classification times. Each classification predicts on a satellite time series from the beginning of the hydrological year 2020 (1 October 2019) until the given date in the image panels.

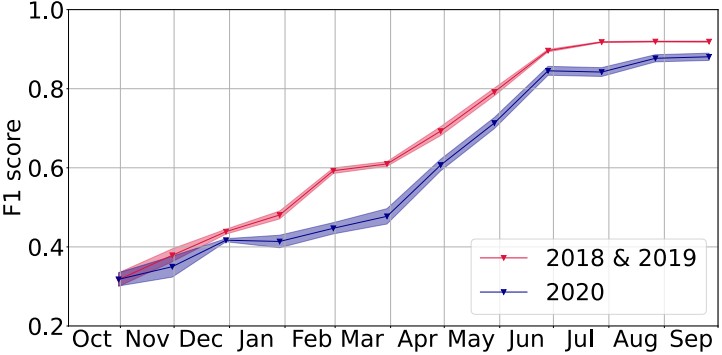

**Figure 7.** The main F1 scores for the unified early classification models trained using 2018 & 2019 data from BB + MV. Each model was evaluated using several masked time series (data acquired until a certain day) on the BB + MV test sets from 2018 & 2019 (red) and the new year 2020 (blue), which was not available during training. The colored bands represent the standard deviation of the scores over the 5 models trained using the same configuration.

Figure 7 displays the main F1 of the (U) model when tested on both 2018 and 2019 and 2020. The metrics for evaluating on the training years are higher than the metrics obtained from inference on the unseen year. This difference is most apparent between January and March. It reaches its maximum difference in the main F1 score of 0.2 at the end of February and then starts decreasing to a 0.05 difference in the main F1 score after the end of June. The main F1 scores of both experiments stabilize by August. At this point the main F1 score of evaluating on the training years has reached the value 0.92 while testing on the unseen year yields a value of 0.84. These closely match the results of the corresponding full year experiments (Section 4.2.2). By the end of June the main F1 scores are 0.90 and 0.85 for the training and unseen years, respectively. This amounts to a difference of results between early and full year classifications of about 0.02 and 0.04.

The corresponding F1 scores for the main crops, along with their approximate seeding and harvesting dates are shown in Figure 8. Most crops reach an F1 score above 0.8 when tested with data that include acquisitions up to the beginning of June. The exceptions to this being potatoes and sugar beet. Potatoes never actually reach 0.8 F1 score and exhibit a consistently wider error bar than the other crops. This relative lower performance for potatoes can also be observed in the full year model (Table 3). We also observe a substantial error bar for sugar beet, and for winter barley and wheat before the end of June. Conversely the metrics for rapeseed never fall bellow the 0.8 F1 score threshold. All of the crop types show an improvement in performance on the period between seeding and harvesting.

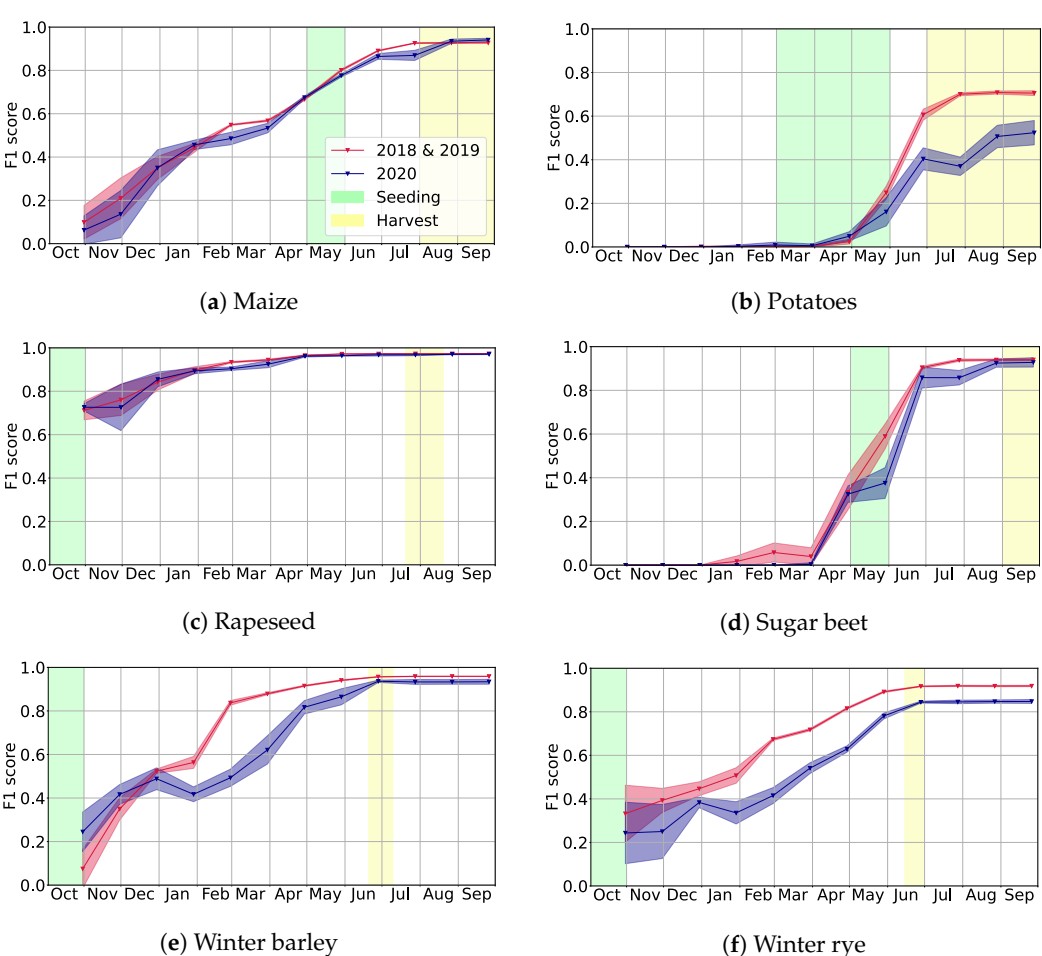

(a) Maize　　　　　　　　　　　　　　　　　(b) Potatoes

(c) Rapeseed　　　　　　　　　　　　　　　　(d) Sugar beet

(e) Winter barley　　　　　　　　　　　　　　(f) Winter rye

**Figure 8.** *Cont.*

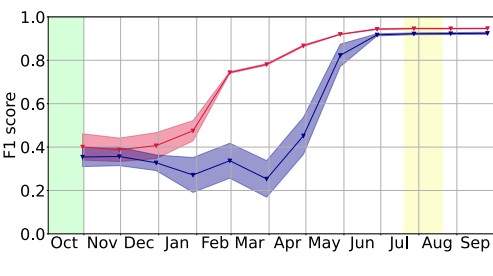

(**g**) Winter wheat

**Figure 8.** Average F1 score for each main crop type when evaluating the unified model (U) using data until the end of the month on the horizontal axis for the years 2018 and 2019 and 2020. The approximate seeding and harvest periods are marked in green and yellow, respectively. The colored bands represent the standard deviation of the scores over five training runs. All models were trained on data from the years 2018 and 2019 from BB + MV.

## 5. Discussion

### 5.1. Full Year Classification

In general, our experiments regarding full year classification, i.e., crop type classification using data from the complete growing season, show that a deep learning approach outperforms classical approaches. The Pixel-Set Encoder–Temporal-Attention Encoder (PSE-TAE) algorithm shows more favorable results than a random forest (RF), when trained and evaluated on the same datasets (main F1 score of 0.91 for PSE-TAE versus 0.72 for RF, see Section 4.1). This finding is in accordance with the results in [18–20,22,23,40]. One important fact here is that our study uses orders of magnitude more data to train and evaluate the different models than all these studies. This circumstance makes it even more favorable to work with a deep learning model, as its iterative training process is inherently suited to cope with large amounts of data. It should be noted that there has been put more effort into optimizing the deep learning approach, since we made the decision to focus only on the better performing algorithm. Therefore, there might still be some potential to improve the RF, but the initial scores do not suggest that the performance could reach the performance of the deep learning approach.

The crop class "potatoes" consistently reaches worse classification performance than all other main crops for both PSE-TAE and RF. This probably has multiple reasons. Not only are there relatively few examples of potato parcels in our datasets, but the crop class potatoes typically encompasses different potato types, which have vastly different seeding time points and growing cycles. This makes this crop type hard to detect for the algorithm.

#### 5.1.1. Data Fusion

Our method for data fusion enables to train models which perform better than the pure Sentinel-1 (S1) or Sentinel-2 (S2) model. The gains compared to using purely S2 data were highest for the classes with few training examples—maybe cloud coverage in the training data obstructed the successful training of the pure S2 model. However, the classification performance when using only S1 or S2 is still almost as high, as when using both. This is valuable information, as it allows to choose the data sources depending on the use case: as using only one of these data sources significantly reduces the amount of data processing needed, one could also opt to use only S1 or S2 with effectively no changes needed in the architecture. Using only S2 would have the advantage of reaching almost the same performance as when using both modalities, especially when one is mostly interested in frequent crop types. On the other hand, using only S1 still leads to very good classification performance, while not being dependent on the availability of cloud free images. Especially in situations where the stability and availability of predictions is more important than reaching the best possible performance or in regions with many clouds, this might be a viable option.

Comparing classification performance to others is only possible in a rather qualitative way, since most publications use datasets differing in region, year, and size. Projects such as [41] provide standardized crop-type data for multiple countries and years in Europe and will hopefully facilitate this task in the future. As [41] does not yet contain data for multiple years in Germany, we were not able to use it for our experiments. All following comparisons of performance metrics with other work are, therefore, to be read with caution:

The way we do data fusion (i.e., early fusion) is also studied by others. Ref. [24] compares different data fusion types. Early fusion reaches the best F1 score in their experiments, although they only report minor differences in performance when using different fusion types. However, their reported F1 score is lower than ours, even when comparing with our evaluations on an unseen year. They use a smaller dataset from a different region. Ref. [25] tries many types of fusion. While early fusion is not one of the best performing methods in their study, they state that it is the most lightweight. Both [24,25] perform temporal interpolation of their satellite time series prior to data fusion, in order to align the S1 and S2 time series. This additional interpolation step makes their fusion method more computation intensive than our method, as we can rely on the positional encoding method to cope with (satellite) time series of different length and cadence, which does not need any temporal interpolation or sampling. This means that our approach allows for the seamless integration of additional data sources (e.g., weather or soil data, such as precipitation or temperature for each day), while saving computational resources for data processing at the same time. This way of providing time information to our model is, therefore, also key to the ease with which we can apply and train our models in other regions and years, since we do not need to rely on temporal interpolation even for vastly different acquisition patterns.

### 5.1.2. Temporal and Spatial Transfer

Temporal transfer, i.e., training a model on different years than evaluating it on, is currently not performed very often, judging from the available literature. For example, Refs. [24,25] train and evaluate their models on the same years and regions. This is useful for comparing different architectures or fusion strategies, but it does not allow to measure performance in a real world application, where one needs to predict crop types in the current year, when there is only training data from previous years available. Ref. [42] do temporal transfer by training on two years and evaluating on a third using a support vector machine and a random forest (RF). We reach higher metrics in terms of Precision and Recall on the common crop types, except for potatoes and Winter Rye, where they have relatively many samples, compared to our dataset. Note that they use a smaller dataset from a different region and different years.

When evaluating on 2020 data, Section 4.2 showed that the model trained on the same year (2020) performed best (main F1 score 0.92). This is not surprising because weather conditions and, hence, growing cycles differ between years. Training on 2018 and 2019 data gave a significantly better model for 2020 evaluation (main F1 score 0.89) than only using a single year (2018 or 2019) for training (main F1 score 0.81 and 0.86, respectively). We would, therefore, expect that adding even more training years can result in a model which can cope even better with weather conditions changing between years—until some saturation is reached. Here, adding weather data to the input might also help, for example by adding temperature and precipitation to the input channels or in the positional encoding. However, we could not find a significant improvement when running preliminary experiments using precipitation and temperature as additional inputs.

We also showed that spatial transfer is feasible—at least when the distance is rather small as from Brandenburg (BB) to Thuringia (TH). Ref. [43] also do spatial transfer between different regions with an architecture that is as well directly derived from [22]. They observe a drop in their Overall Accuracy in the same order of magnitude of the drop we observe in our main F1 score, although they use different datasets from ours. Ref. [44] do spatial transfer between Germany and South Africa using RF and convolutional neural

networks (CNN). They report lower F1-Scores than our model, but the two regions in this study have extremely different climate and cropping patterns.

As expected, training on the evaluation region still leads to the best results. However, performance in the spatial transfer task can be improved by using data from more regions: Training on Brandenburg and Mecklenburg-Vorpommern (BB + MV) and evaluating on TH only led to a decrease of 0.02 in main F1 score compared with a model trained on TH. This further underlines the general notion already observed for temporal transfer: using more diverse data for training improves performance on data from new regions, as the model's capacity seems to be large enough to learn from these different conditions simultaneously. Therefore, one idea for future research would be to utilize even more diverse training data from different climate and with different cropping patterns to allow for a wider applicability of the model. These efforts might be supported by additionally providing weather information or elevation data.

### 5.2. Early Classification

Our proposed early classification strategy of training a unified model on 2018 & 2019 data yields a good predictive performance during 2020. As one can see in Figure 7, it reached a main F1 score over 0.8 using the data available at the end of June 2020. Interestingly, non-dominant crop phenologies (e.g., rapeseed flowering) could be identified, leading to a high increase in the classification accuracy.

While testing our early classification models on data from 2020 (Figure 5b) a dip in performance in July is apparent. We noticed a drop in the average number of measurements from S2, relative to the training years, within this period and assume this to be a probable reason for the reduction in performance by the model as it would expect more information from the optical sensors for this time. A possible way to cope with this in the future is to include time series with similar behavior so as to prevent the model from relying on the presence of these data points. One could even go one step further and randomly delete S2 acquisitions from the training time series, as a form of data augmentation. This could make the model even more robust to missing S2 data. An interesting aspect here is that the unified model (U) has a less pronounced reduction and variation in performance at the same time point. This leads us to believe that it is able to cope better with an unexpected lack of data than the separate model (S).

Early crop classification has been studied by others as well. Comparison of performance can again only be completed in a rather qualitative way. Refs. [36,42,44–47] do early crop classification using classical machine learning algorithms (random forests, support vector machines or bagged trees), Ref. [44] also experiments with a CNN-based architecture. The performance reported there is mostly lower or on par with our results, except for maize, which sometimes reaches slightly higher classification scores than our results, especially early in the year. Due to the different databases it is hard to say whether this is due to an advantage of the algorithms used or a property of the dataset. One of the studies which reaches a high performance for maize early in the season is [47]. In addition to S1 data, they also use data from a Digital Elevation Model (DEM). This kind of data could be integrated into our model as well and their results indicate that this could be an interesting aspect for future research. In addition, only [42,47] evaluate their algorithm in a different year from the one it was trained in, which would be the actual use case, especially for an early classification algorithm.

Ref. [48] use a decision tree devised by experts. This means that a parcel is classified as a certain crop based on a known property being apparent on a certain period of time. They develop their method for the year 2015 and then apply it to the year 2016. The adjustment to 2016 needs again some manual input from experts. With this in mind, we can still detect Barley and Rye at least two months earlier than this study (with an F1 score > 0.8 for the evaluation in an unseen year), Rapeseed and Sugar Beet at least a month earlier. According to the study, maize seems to start being distinguishable in mid June, this corroborates our

model only reaching peak performance in this month. The only crop which our model can reliably detect later is Wheat, which [48] detect from April while our model does so in May.

The authors of [24] rerun their experiment for early classification. In this research, fusing S1 and S2 (using early fusion similar to ours) slightly outperformed using only S2 data. Ref. [49] uses an LSTM network with an altered loss function that takes into account a probability of stopping. This means that the model has a decision mechanism that, after "seeing" some amount of data, decides whether the prediction is good enough (the algorithm "stops") or that it needs to see more of the time series to produce a better prediction. They only use the year of 2018 for training and evaluation and only S2 data. Interestingly, their stopping times somewhat coincide with the time our prediction stabilizes (when looking at the standard deviation from multiple trainings) for the crops we have in common: for Winter Wheat, most of their classifications stop between June and August, our approach stabilizes by the end of June. Similar observations hold for Maize and Barley. Their maximum F1 score is considerably below ours.

All works mentioned above use separate models for each time point in early classification. The only exception here being [46], but they depend on having enough cloud free images available, which is a significant drawback, especially in the early season in Europe. Our unified model has two main advantages compared to this separate model approach: the training times are shorter since we only need to train a single model. Additionally, we can make predictions at any point in the year. Any time series containing observations between October 1 of the preceding year and some day within a 366 day interval can be used as input. It could be shown that the classification performance of this approach is equivalent to a separate model approach in our setup, with the potential to be even more stable with regard to missing S2 data.

## 6. Conclusions

In this article, we presented a flexible deep learning based method for the classification of crop types growing on a given parcel. The corresponding model performs best when using Sentinel-1 and Sentinel-2 data together, while also leading to reasonable performance when only relying on data from Sentinel-1. We showed that it generalizes well to a new year that was not seen during training. Furthermore, we observed that using training data from several years and regions simultaneously increases prediction performance and that using additional train and validation data should be helpful for future research to analyze the generalization abilities of the approach. For both aspects, one can imagine some saturation, for example when lots of different weather conditions were used in the training years. Crucially, we may conclude that our model can conduct classification early in the year: we typically reached F1 scores above 0.8 at least a month before harvest time.

Apart from the amount of data it was trained on, the two main advantages of our model are that on the one hand it can easily incorporate further data sources without the need for temporal interpolation due to its flexible data fusion mechanism, and on the other hand that a single model can predict crops at an arbitrary time point during the growing season.

In order to scale the application of the model (for example to be able to operationalize it as a service), the transferability to other regions with different growing patterns, smaller field sizes, and different crop types need to be studied.

**Author Contributions:** Conceptualization, D.S. and J.W.; data curation, R.B., R.G., A.M., T.S. and F.W.; methodology, D.S. and J.W.; writing, F.W., R.B., L.R., T.S., D.S. and J.W.; supervision, L.R., D.S. and J.W. All authors have read and agreed to the published version of the manuscript.

**Funding:** This research was funded by the "Central Innovation Programme for small and medium-sized enterprises (ZIM)" of the German Federal Ministry for Economic Affairs and Climate Action under grant number 16KN089024.

**Institutional Review Board Statement:** Not applicable.

**Informed Consent Statement:** Not applicable.

**Data Availability Statement:** The Copernicus Sentinel-1 and Sentinel-2 datasets used in this study are freely and openly available. Parcel label data for Brandenburg (BB) can be downloaded from [34]. Parcel label data for Mecklenburg-Vorpommern (MV) and Thuringia (TH) was provided to the authors by public authorities and can be requested from "Ministerium für Landwirtschaft und Umwelt Mecklenburg-Vorpommern" (for MV) and from "Thüringer Landesamt für Landwirtschaft und Ländlichen Raum" (for TH), respectively.

**Conflicts of Interest:** The authors declare no conflict of interest.

**Abbreviations**

The following abbreviations are used in this manuscript:

| | |
|---|---|
| BB | Brandenburg |
| CNN | Convolutional neural network |
| DEM | Digital elevation model |
| DL | Deep learning |
| MV | Mecklenburg-Vorpommern |
| NDVI | Normalized difference vegetation index |
| NDWI | Normalized difference water index |
| PSE | Pixel-set encoder |
| RF | Random forest |
| S1 | Sentinel-1 |
| S2 | Sentinel-2 |
| SAR | Synthetic aperture radar |
| TAE | Temporal-attention encoder |
| TH | Thuringia |
| VH | Vertical horizontal Sentinel-1 polarization |
| VV | Vertical vertical Sentinel-1 polarization |

**Appendix A. Details on the Model Architecture**

For our crop classification algorithm we build on the Pixel-Set Encoder–Temporal-Attention Encoder (PSE-TAE) model architecture from [22], incorporating some slight modifications which we shall detail in the following. Here, we give some details illuminating Section 3.1.

*Appendix A.1. Pixel-Set Encoder*

The data from each parcel $p$ are given as an array (or tensor) of shape $T_p \times C \times H_p \times W_p$, where $T_p$ is the number of time points with observations for this parcel, $C$ the number of channels, $H_p$ and $W_p$ are the height and width of the observations. The pixels outside the parcels have values NaN. Additionally, we have a list $(\text{day}(1), \ldots, \text{day}(T_p))$ of the days of the data acquisitions, where $\text{day}(t)$ is the number of days of the $t$-th observation after October 1st. The time dimension of the parcel data and the list of days of observations are padded to length $T = 140$. The parcel data then have the shape $T \times C \times H_p \times W_p$ for each parcel and the list of days of acquisitions always has length $T$, regardless of the actual number of available acquisitions. The value of $T = 140$ was picked by evaluating the number of available acquisition time points in our training dataset, leaving a generous margin to allow for other parcel sets having more acquisition time points available within one year.

We sample a set $S$ of pixels from each parcel ($|S| = 10$), allowing repetition of pixels. The observation in each pixel $s$ at time $t$ ($1 \leq t \leq T$) is a vector of length $C$, to which we apply two fully connected layers, each having an output dimension of 32, from which we obtain a vector $\hat{e}_s^{(t)}$ of length 32 for each $s \in S$.

Then we apply a fully connected layer $f$ to the 64-dimensional concatenation of the mean and standard deviation over $S$:

$$\{\widehat{e}_s^{(t)}\}_{s\in S} \mapsto f\left(\begin{array}{c} \mathrm{mean}_{s\in S}\,\widehat{e}_s^{(t)} \\ \mathrm{std}_{s\in S}\,\widehat{e}_s^{(t)} \end{array}\right) =: e^{(t)} \in \mathbb{R}^{d_e}.$$

The vector $e^{(t)}$ encodes the $t$-th observation—we choose its dimension to be $d_e = 128$. We do not use geometrical features as suggested in [22], mainly for simplicity. Overall, we concatenate the resulting embeddings of the observations into the time series $e = [e^{(1)}, \ldots, e^{(T)}]$ of shape $128 \times T$.

*Appendix A.2. Positional Encoding*

We add a positional encoding $p$ ("positional" referring to a temporal position here) to the pixel-set encoding $e$: We send the time series $(e^{(1)} + p^{(1)}, \ldots, e^{(T)} + p^{(T)})$ to the TAE, where

$$p_i^{(t)} = \sin\left(\mathrm{day}(t)/\tau^{\frac{2i}{d_e}} + \frac{\pi}{2}(i \mod 2)\right)$$

for $1 \leq i \leq d_e$. We kept using $\tau = 1000$. Note that the acquisition days between the parcels vary, i.e., the $t$-th observation from some parcel is usually from a different day than the $t$-th observation from some other parcel, cf. Section 2. Therefore, the use of $\mathrm{day}(t)$ in $p$ is crucial, as it allows us to define a temporal position for each observation, independent of its position $i$ in the parcel data array. The difference between the days becomes rather large when one of these parcels has a longer time interval without observations due to clouds (when using Sentinel-2). One might interpolate the missing observation from an early and a later observation. We decided to use the pure input data in contrast to [22], because this way, we do not need to artificially create observations by temporal interpolation.

*Appendix A.3. Temporal-Attention Encoder*

The original PSE-TAE architecture [22] uses a slightly modified self-attention layer, where (for each attention head) one first applies a fully connected layer to $e^{(t)} + p^{(t)}$ to compute the key and query vectors and then takes the mean of the query vectors over time $t$ to obtain a "master query". We also use this architecture, but apply some modifications:

- We first apply a conventional self-attention layer to $e + p$ before using the layer with the master query.
- We are using 8 attention heads instead of 4.
- We also apply a fully connected layer to $e + p$ to obtain the value vectors as proposed in [43].

The first two modifications slightly stabilized validation loss curves during training, the third modification decreased the number of parameters and hence the training time.

After concatenating the outputs from the 8 attention heads, we apply two fully connected layers with hidden dimension 128 to obtain the encoding of all the information from the parcel. Three consecutive fully connected layers with hidden dimensions 128, 64, and 32 and 17-dimensional output are then applied to compute the logits of the crop types.

## Appendix B. Details on Random Forest Baseline

Similarly to [22], for our baseline model we implemented a random forest (RF) following the unstructured configuration of [16], opting for only including data from Sentinel-2. Following the idea from [39] we decided to use bands 3 to 8 plus bands 11 and 12, as well as the Normalized Difference Vegetation Index (NDVI), the Normalized Difference Water Index (NDWI), and the brightness (Euclidean norm of the surface reflectances). In order to deal with the missing values caused by clouds we linearly interpolated the missing data points for the existing observations time-wise. After that the mean and standard deviations for each feature (bands, indices, and brightness) were computed parcel-wise. To keep all

time series the same size, we padded the existing data points with zeros summing up to 140. This resulted in a time series of 140 time steps containing $2 \times 13$ features (mean and standard deviation for each band, index and brightness). Finally, the time series for each parcel were stacked into a two-dimensional array $X$ were each column contains the value of a feature at each time step. A one-dimensional array $y$ with length equal to the number of rows from $X$ was also generated, containing the labels of the parcels. A RF with 500 trees was then trained and tested. We used weighted sampling to address the unbalanced distribution of crop types while training. To serve as comparison, we trained a deep learning model in a similar way as the models presented in Section 4.2. Both models used the same training and testing sets obtained from data from Brandenburg and Mecklenburg-Vorpommern (BB + MV) from 2018. These splits were created as described in Section 3.5. Table A1 shows the results of testing the RF on the test set of the same dataset used for training.

**Table A1.** Evaluation of the RF model trained on BB + MV 2018, detailed by main crop type. The evaluation is performed on the test set from the training dataset.

|              | Recall | Precision | F1   | Support |
|--------------|--------|-----------|------|---------|
| Maize        | 0.94   | 0.62      | 0.75 | 1741    |
| Potatoes     | 0.29   | 0.34      | 0.32 | 140     |
| Rapeseed     | 0.96   | 0.82      | 0.89 | 992     |
| Sugar beet   | 0.79   | 0.80      | 0.80 | 128     |
| Winter barley| 0.81   | 0.69      | 0.74 | 812     |
| Winter rye   | 0.74   | 0.54      | 0.63 | 1385    |
| Winter wheat | 0.74   | 0.70      | 0.72 | 1687    |
| Other        | 0.90   | 0.98      | 0.94 | 21,544  |
| Average\|Sum | 0.77   | 0.69      | 0.72 | 28,429  |

**Appendix C. Details on Model Training**

We implemented the experiments presented here in PyTorch. We trained the models on an Nvidia GeForce RTX 3090 GPU. The CPU we used has 64 physical cores, running 128 threads in parallel. This high number of threads was particularly useful because we needed to load the data from lots of parcels in parallel.

As a loss function, we used cross-entropy loss for the 17 classes. In order to minimize it, we applied the Adam optimizer [50] with weight decay of 0.01 and the usual choice of $\beta = (0.9, 0.999)$. The batch size used was 32. To schedule the learning rate, we used linear warm-up cosine annealing with a maximum learning rate of 0.0002. We usually trained for 20 to 30 epochs, then selected the model having the lowest loss value on the validation set (from the same year(s) and state(s)). The number of necessary epochs varies because the size of the training set varies, e.g., we typically needed less epochs to train when we used two states instead of one. The model has about 400,000 trainable parameters and takes 2 GB of graphics RAM. Training a single model using data for a full year typically took between a few hours to one day, depending on the size of the training dataset.

**Appendix D. Model Ablation: TAE without PSE**

Here we investigate the usefulness of the Pixel-Set Encoder (PSE) as described in Section 3.1. We train two Temporal-Attention Encoder (TAE) models, one with the PSE (PSE-TAE) and the other without it. For the model without the PSE we just compute the band-wise average of 10 randomly selected parcel pixels (AVG-TAE). Both models are trained and tested on Brandenburg and Mecklenburg-Vorpommern (BB + MV) in years 2018 and 2019 (2018 & 2019). The results show that the model performs better when using PSE as a spatial encoder (Table A2). This is in accordance with [22].

**Table A2.** Main and overall F1 scores for full year inference on test data using PSE or average of randomly selected pixels as spatial encoders. Both models were trained and tested using data from BB + MV in 2018 & 2019, only using S2 observations.

|  | **Main F1** | **Overall F1** |
|---|---|---|
| AVG-TAE | 0.86 | 0.67 |
| PSE-TAE | 0.92 | 0.81 |

## Appendix E. Detailed Metrics

**Table A3.** Detailed version of Table 2 in Section 4.2.1. For each crop type and modality (S1, S2, or both), we show the average F1 score over the 5 trained models. The models were trained and tested on Brandenburg 2018. Note that the support here counts the number of parcels in the test set, the train set is approximately 7 times larger (Section 3.5).

|  | **S1** | **S2** | **S1 + S2** | **Support** |
|---|---|---|---|---|
| Maize | 0.81 | 0.85 | 0.85 | 865 |
| Potatoes | 0.49 | 0.54 | 0.60 | 66 |
| Rapeseed | 0.94 | 0.97 | 0.97 | 433 |
| Sugar beet | 0.86 | 0.94 | 0.94 | 40 |
| Winter barley | 0.87 | 0.93 | 0.94 | 391 |
| Winter rye | 0.88 | 0.91 | 0.92 | 1076 |
| Winter wheat | 0.86 | 0.87 | 0.91 | 634 |
| Asparagus | 0.52 | 0.64 | 0.66 | 11 |
| Lupins | 0.53 | 0.63 | 0.69 | 80 |
| Meadows | 0.83 | 0.88 | 0.89 | 5909 |
| Peas | 0.80 | 0.75 | 0.82 | 46 |
| Spring barley | 0.44 | 0.55 | 0.62 | 70 |
| Spring oats | 0.45 | 0.59 | 0.62 | 213 |
| Spring rye | 0.22 | 0.31 | 0.47 | 49 |
| Spring wheat | 0.31 | 0.37 | 0.42 | 55 |
| Sunflower | 0.67 | 0.73 | 0.79 | 63 |
| Not considered | 0.79 | 0.84 | 0.84 | 6223 |
| Average | Sum | 0.66 | 0.72 | 0.76 | 16,224 |

**Table A4.** Detailed version of Table 3. Evaluation of full year classification for all crop types on test set for Brandenburg and Mecklenburg-Vorpommern (BB + MV) from 2018 & 2019. The model was trained on the training set for BB + MV from 2018 & 2019. Note that the support here counts the number of parcels in the test set, the train set is approximately 7 times larger (Section 3.5).

|  | **Recall** | **Precision** | **F1** | **Support** |
|---|---|---|---|---|
| Maize | 0.94 | 0.91 | 0.93 | 3672 |
| Potatoes | 0.67 | 0.82 | 0.74 | 267 |
| Rapeseed | 0.97 | 0.98 | 0.97 | 1760 |
| Sugar beet | 0.94 | 0.95 | 0.94 | 244 |
| Winter barley | 0.96 | 0.95 | 0.96 | 1638 |
| Winter rye | 0.92 | 0.92 | 0.92 | 3052 |
| Winter wheat | 0.93 | 0.95 | 0.94 | 3489 |
| Asparagus | 0.59 | 0.85 | 0.69 | 29 |
| Lupins | 0.70 | 0.84 | 0.76 | 257 |
| Meadows | 0.91 | 0.90 | 0.90 | 20,642 |
| Peas | 0.88 | 0.82 | 0.85 | 237 |
| Spring barley | 0.73 | 0.74 | 0.74 | 313 |
| Spring oats | 0.65 | 0.72 | 0.68 | 663 |
| Spring rye | 0.52 | 0.55 | 0.54 | 121 |
| Spring wheat | 0.49 | 0.62 | 0.55 | 174 |
| Sunflower | 0.75 | 0.95 | 0.84 | 154 |
| Not considered | 0.86 | 0.86 | 0.86 | 19,864 |
| Average | Sum | 0.79 | 0.84 | 0.81 | 56,576 |

**Table A5.** Detailed metrics for temporal transfer (Table 3). Evaluation on test set for BB + MV from 2020. Training was done on training data set for BB + MV from 2018 & 2019. Note that the support here counts the number of parcels in the test set, the train set is approximately 7 times larger (Section 3.5).

|  | Recall | Precision | F1 | Support |
|---|---|---|---|---|
| Maize | 0.94 | 0.95 | 0.95 | 2035 |
| Potatoes | 0.59 | 0.65 | 0.62 | 131 |
| Rapeseed | 0.96 | 0.98 | 0.97 | 745 |
| Sugar beet | 0.91 | 0.97 | 0.94 | 118 |
| Winter barley | 0.85 | 0.97 | 0.91 | 858 |
| Winter rye | 0.86 | 0.78 | 0.82 | 1607 |
| Winter wheat | 0.90 | 0.93 | 0.91 | 1488 |
| Asparagus | 0.67 | 0.33 | 0.44 | 6 |
| Lupins | 0.65 | 0.65 | 0.65 | 107 |
| Meadows | 0.83 | 0.91 | 0.87 | 10,609 |
| Peas | 0.75 | 0.94 | 0.84 | 142 |
| Spring barley | 0.52 | 0.44 | 0.48 | 99 |
| Spring oats | 0.38 | 0.73 | 0.50 | 370 |
| Spring rye | 0.43 | 0.32 | 0.37 | 88 |
| Spring wheat | 0.17 | 0.20 | 0.18 | 46 |
| Sunflower | 0.58 | 0.81 | 0.68 | 86 |
| Not considered | 0.87 | 0.79 | 0.83 | 10,649 |
| Average \| Sum | 0.70 | 0.73 | 0.70 | 29,184 |

**Table A6.** Detailed main F1 scores used in Figure 7. These values represent the averages along with the standard deviation resulting from testing five (U) models trained on BB + MV 2018 & 2019. Each model is tested for both the training years (2018 & 2019) and the unseen year (2020). To obtain a prediction at each point in time, temporal masking, as described in Section 3.4, is used. Each prediction is computed from data that were observed between the 1 October up to its corresponding point in time throughout the hydrological year.

|  | Test Year(s) | |
|---|---|---|
|  | **2018 & 2019** | **2020** |
| 31 October | $0.32 \pm 0.02$ | $0.32 \pm 0.02$ |
| 30 November | $0.38 \pm 0.02$ | $0.35 \pm 0.03$ |
| 30 December | $0.44 \pm 0.01$ | $0.42 \pm 0.004$ |
| 29 January | $0.48 \pm 0.01$ | $0.41 \pm 0.02$ |
| 28 February | $0.59 \pm 0.01$ | $0.45 \pm 0.01$ |
| 30 March | $0.61 \pm 0.01$ | $0.48 \pm 0.02$ |
| 29 April | $0.69 \pm 0.01$ | $0.61 \pm 0.01$ |
| 29 May | $0.79 \pm 0.01$ | $0.71 \pm 0.01$ |
| 28 June | $0.90 \pm 0.003$ | $0.85 \pm 0.01$ |
| 28 July | $0.92 \pm 0.001$ | $0.84 \pm 0.01$ |
| 27 August | $0.92 \pm 0.001$ | $0.88 \pm 0.01$ |
| 26 September | $0.92 \pm 0.002$ | $0.88 \pm 0.01$ |

**Appendix F. Crop Type Mappings**

As described in Section 2.2, the datasets containing crop types for the states Brandenburg (BB), Mecklenburg-Vorpommern (MV), and Thuringia (TH) needed to be harmonized to the same crop type definitions from the Joint Experiment for Crop Assessment and Monitoring (JECAM) [35]. The corresponding mappings between state specific crop code and JECAM code can be found in Table A7.

**Table A7.** Mapping between state specific crop codes for Brandenburg (BB), Mecklenburg-Vorpommern (MV), Thuringia (TH), and JECAM codes. Used for harmonization of crop types.

| Crop Name | JECAM Code | BB Crop Code(s) | MV Crop Code(s) | TH Crop Code(s) |
|---|---|---|---|---|
| Maize | 12 | 171, 172, 411 | 171, 172, 411 | 411,000, 171,000, 411,009 |
| Potatoes | 51 | 601, 602, 606 | 601, 602, 606 | 610,110 |
| Lupins | 76 | 230 | 230 | 230,010 |
| Peas | 77 | 210 | 210, 240 | 210,000 |
| Sugar beet | 81 | 603 | 603 | 620,000 |
| Meadows | 91 | 422, 424, 433, 451, 452, 453, 459 | 422, 424, 433, 451, 452, 453 | 418,200, 418,360, 428,000, 451,000, 452,000, 453,000, 500,020 |
| Winter wheat | 111 | 115 | 112, 115, 432 | 111,150, 190,130 |
| Spring wheat | 112 | 116, 851 | 113, 116 | 111,160 |
| Winter barley | 151 | 131 | 131 | 190,310 |
| Spring barley | 152 | 132 | 132 | 190,320 |
| Winter rye | 161 | 121 | 121 | 190,210 |
| Spring rye | 162 | 122 | 122 | 190,220 |
| Spring oats | 172 | 143 | 143 | 190,410 |
| Asparagus | 212 | 8, 860 | 860 | 717,150 |
| Rapeseed | 435 | 311, 312 | 311, 312 | 311,000 |
| Sunflower | 438 | 320 | 320 | 320,000 |

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
