# Peer review of "Early Crop Classification via Multi-Modal Satellite Data Fusion and Temporal Attention"

_remotesensing, doi:10.3390/rs15030799_

Round 1

Reviewer 1 Report

The paper proposes a deep learning-based algorithm for classifying crop types from Sentinel-1 and Sentinel-2 time series data. The experimental results demonstrate the approach's effectiveness on an extensive data set from three federal states of Germany. The contribution of this paper is prominent, but some parts should be improved.

1. In the Abstract, line 5 indicates, "Such reliable early season predictions..." Why it is a reliable prediction, you should propose sufficient evidence.

2. There are too many keywords.

3. The background and definition of the problems should be improved in the Introduction; the purpose and innovation of the present study is not very accurate and concise. I think the motivation is early crop classification, which is not mentioned in the title.

4. In the Introduction section, you should state why this method is chosen. For example, why do you choose multi-modal satellite data fusion and temporal attention?

5. In the Data section, you should describe why you select such research areas.

6. In describing the method, I recommend emphasizing your innovation.

7. In the Discussion section, it's better to present the principles and generalizations shown by the experimental results. Besides, I suggest showing how your results and interpretations agree or contrast with the pertinent literature.

8. In the Conclusion section, it's better to point out any exceptions or unsettled points.

Reviewer 2 Report

This article describes a deep learning-based algorithm for classifying crop types from time series data collected by Sentinel-1 and Sentinel-2 satellites. The algorithm is based on the transformer architecture and is able to predict crop types at the early stages of the growing season with a single trained model, which can be useful for forecasting yields or modeling agricultural water usage.

Although I see the work as positive in general, I request some parts that I mentioned below be corrected.

1) First of all, the majority of the rest of the part between lines 18-29 in the introduction part is taken from another study and the similarity rate of this part should be changed.

2) Likewise, the part between lines 146-159 of the method part was written from another article. Similarities in this large cannot be accepted, even if they are your own work.

3) I recommend that the challenges (line 202) section given under the title of Method should be transferred to the discussion section.

4) In terms of written language, the pronoun "we" is used too much. Instead, passive sentences should be used.

5) A visual map of the classifications should be added.

Reviewer 3 Report

This manuscript describes an efficient classification method to identify early crop with Sentinel-1 and Sentinel-2 time series data. This is a relevant topic making this a valuable manuscript. However, I still have some comments before the manuscript can be acceptted.

1. How many ground truth samples were used? Maybe only frequencies of crop types in Figure 3 is not enough. Please add a figure to present the location of ground truth sites.

2. In addition to Figure 4, the general framework of early crop classification is needed to clarify the procedure of multi-modal.

3. In Section 4 and 5, classification results, including class metrics of different crop types, classified map, should be added.

Reviewer 4 Report

The manuscript intends to propose a deep learning algorithm for early crop classification. The current manuscript should be improved by well re-organizing, especially for the introduction, method and results. Several parts of the manuscript look more like an experiment report rather than a prepared publication. They should be improved. Detail comments are as follows:

1.      Please improve the introduction by adding current challenge in early crop classification and the highlight.

2.      What is the emphasis and challenge for early crop estimation? These issues should be highlighted in the introduction.

3.      I would suggest delete the subtitle as Outline of the article. I think it is not necessary.

4.      Figure 2: What is the size of the parcels? Are they in 10km*10km tiles? Aren’t they related with the field size?

5.      Figure 3: I did not understand what does the frequency of crop type represent for? Especially the value of the y axis? For example, what does a value of 104 illustrate for maize frequency? I think it should be explained.

6.   The section of method and the results can be improved and re-organized. The current version is mixed with background, challenge and the description of experiments thoughts. Some sentences are not necessary, such as the first sentence in Results.

7.      How does the data from S1 and S2 fuse? What are the date of these data? How does it implement for early crop identifying?

8.      How does other accuracy criterion performance? Is the result reliable with the only evaluation metrics as F1? And is it reliable across years?

9.      Figure 5: Is the evaluation result related to dominant crop phenology? It should be discussed.

Round 2

Reviewer 2 Report

The authors made the desired edits. It can be published as is.

Reviewer 4 Report

The authors has addressed all the comments and revised the manuscript accordingly.